# A New Blockchain-Based Multi-Level Location Secure Sharing Scheme

**Qiuhua Wang** [1,*] **, Tianyu Xia** [1] **, Yizhi Ren** [1] **, Lifeng Yuan** [1] **and Gongxun Miao** [2]

[1] School of Cyberspace, Hangzhou Dianzi University, Hangzhou 310018, China; 15084234@hdu.edu.cn (T.X.); renyz@hdu.edu.cn (Y.R.); yuanlifeng@hdu.edu.cn (L.Y.)

[2] Zhongfu Information Co., Ltd., Jinan 250101, China; miaogx@zhongfu.net

[*] Correspondence: wangqiuhua@hdu.edu.cn; Tel.: +86-0571-868-73820

**Abstract:** Currently, users' location information is collected to provide better services or research. Using a central server to collect, store and share location information has inevitable defects in terms of security and efficiency. Using the point-to-point sharing method will result in high computation overhead and communication overhead for users, and the data are hard to be verified. To resolve these issues, we propose a new blockchain-based multi-level location secure sharing scheme. In our proposed scheme, the location data are set hierarchically and shared with the requester according to the user's policy, while the received data can be verified. For this, we design a smart contract to implement attribute-based access control and establish an incentive and punishment mechanism. We evaluate the computation overhead and the experimental results show that the computation overhead of our proposed scheme is much lower than that of the existing scheme. Finally, we analyze the performances of our proposed scheme and demonstrate that our proposed scheme is, overall, better than existing schemes.

**Keywords:** location sharing; privacy preserving; blockchain; Merkle tree; smart contract; attribute-based access control





## 1. Introduction

With the development of technologies such as smart phones [1], smart wearable devices [2], and the Internet of Things [3], the location of target objects can be collected for enterprises or governments to meet business needs, such as vehicle navigation [4], epidemic prevention [5] and social software. For example, since December 2019, the COVID-19 virus has spread all over the world. Scientific epidemic prevention and control measures can effectively curb the spread of the epidemic and reduce the population infection rate. For this reason, it is of great significance to grasp the location information of cases, suspected cases, and the close contacts of cases. For this, location-based services (LBS) [6–8] have been developed and popularized.

Location information contains a large amount of private user information, which is a valuable information resource that should be properly managed. In the process of location sharing, if the location data are leaked, tampered with or forged, this will not only reveal the private information but also negatively affect the business of location acquisition services. Especially in a large-scale location-sharing system, containing a large number of location providers and location demanders, how to ensure information security and privacy protection has become a serious problem that cannot be ignored.

Undifferentiated information sharing [9] can no longer meet the needs of user privacy protection in location information sharing; hence, a secure, efficient, multi-level location information-sharing scheme is required. Location data demanders should be allowed to obtain location data with limited accuracy. For example, a social software server should be allowed to obtain a larger location area to determine the city in which the user is located.



However, a healthcare provider should be allowed to obtain a street-level area or even the exact user's location coordinates. This requires multi-level access control for location data.

Traditional data-sharing schemes [10–12] use centralized servers to process and save data, and implement certain access control rules. For the centralized servers, users' privacy cannot be fully protected, even with the help of some privacy protection technologies, such as k-anonymity [13,14], which will cause data loss. If the centralized server is attacked, large-scale data security problems will arise. Moreover, even if many users in the network already have relevant data, the centralized server must also participate in all data-sharing processes, making the burden of the centralized server very high.

There is no centralized node or authoritative manager in the blockchain network. Blockchain is a distributed ledger and, based on the agreed specifications and consensus protocols [15–17], it is jointly maintained by the nodes of the blockchain network in a de-trusted environment. Smart contracts [18,19] support various data-access strategies, such as attribute-based access control (ABAC [20–22], and role-based access control (RBAC) [23–25]. Data can be circulated in the blockchain network according to the custom rules. In addition, the data structure of the blockchain has a strong immutability. All the above properties endow the blockchain technology with an inherent advantage in the field of data sharing.

The following problem needs to be solved: while effectively sharing location information, the user's location privacy must be protected, and the received data must be true and credible.

To solve this problem, in this paper, we propose a new blockchain-based, multi-level secure location sharing scheme. In our proposed scheme, location information is defined as multiple different accuracies. Location information demanders (*LD*s) are assigned different access levels, so they can learn the area where location information providers (*LP*s) are located with different accuracies. Moreover, we ensure that the received data can be verified. Our main contributions are summarized as follows:

(1) We design a new method to strip the location information into multi-level virtual location data and offset vectors, thus realizing the multi-precision setting of location information;

(2) We design a smart contract to implement fine-grained, attribute-based, multi-level access control to the location data in a decentralized environment;

(3) We establish an incentive and punishment mechanism through a smart contract to ensure the honest operation of nodes in the location-sharing system;

(4) We record the root node of the Merkle tree and the hash value of the data in the blockchain to realize the immutability and verifiability of the data;

(5) We evaluate the computation overhead in the proposed scheme, and the experimental results show that our proposed scheme is more effective than the existing scheme;

(6) We analyze the performances of our proposed scheme and prove that it not only inherits the decentralization and immutability of the blockchain but also has the properties of multi-level privacy protection, verifiability, etc. Compared with other existing schemes, our proposed scheme has better performance overall.

The organization of this paper is as follows. Related works are introduced in Section 2. We briefly introduce the smart contract and Merkle tree in Section 3. The system model and the scheme of this paper are described in Section 4. We evaluate the computation overheads of each phase in our proposed scheme in Section 5. We analyze the performances of our proposed scheme in Section 6. Section 7 concludes the paper.

## 2. Related Works

Using a centralized server to share location [9] is a common method. Dodgeball is one of the earliest location-based online social network (LSN) services; founded by Dennies Crowley and Alex Rainert in 2000, it relied heavily on short message services (SMS). Users texted their location to the service, which then notified other users. Google launched the "Latitude" service in 2009. Google latitude allows users to update their locations by manually typing an address, so as to disable real-time tracking and hide their locations

from others. In WhatsApp [26], users are allowed to share their current coordinates to others over the system. However, if users share their location in a group consisting of individuals they have blocked, the blocked individuals will still be able to see their location. During the COVID-19 pandemic, Singapore and Australia used centralized apps [27], and sent data collected by a user's phone to a central database controlled by a national health agency or another governmental authority. In April 2020, Apple [28] and Google revealed that they were busy developing privacy-friendly contact-tracing technology, which is inspired by the ultra-privacy-sensitive DP-3T protocol. The protocol includes: (1) The use of Bluetooth. (2) The generation of random identification numbers by phones that change every 10–20 min. (3) An opt-in system. (4) The data being stored and processed on users' devices.

In 2008, S. Nakamoto [29] proposed blockchain technology to prevent the double-spending problem using a peer-to-peer network. Although the blockchain was originally developed as a trading platform for virtual cryptocurrency, the application of this technology in different fields [30,31] in recent years has been surprising. Among them, blockchain is expected to change the form of data management and sharing in many scenarios.

Guy zyskind et al. [32] proposed a blockchain-based personal data storage scheme UB-PPD. In UB-PPD, two types of transactions, namely, $T_{access}$ for access control management and $T_{data}$ for data storage and retrieval, are recorded in the blockchain as an automatic access controller. The user's personal data, including location data, are stored and accessed off-chain through Distributed Hash Table (DHT). Asaph Azaria et al. [33] proposed a blockchain-based, decentralized record-management system (MedRec) to process Electronic Medical Record (EMRs). Medical data are stored in care-providers' local database. Three types of smart contract, Registrar Contract (RC), Summary Contract (SC), and Patient-Provider Relationship Contract (PPRC), are used to manage the process of access control and data sharing. In the above two works, the requester cannot verify the received data, and the privacy protection of the data recorded in the blockchain is not considered. In UB-PPD, the nodes in the blockchain network can even freely access the user's personal data. Kai Fan et al. [34] proposed a blockchain-based medical-information-sharing scheme, Medblock. EMRs are stored in external databases, and the ciphertext and the summary information containing the storage address of the EMR are recorded in the blockchain. Any user can access the hash value in the blockchain to verify the data, while the access control protocol recorded in the block header is executed to allow legitimate users to obtain the ciphertext of the summary. Unfortunately, the access control in Medblock is inefficient. Blockchain, as a distributed database, is not suitable for recording heavyweight data.

For location data, Yaxian Ji et al. [35] proposed a blockchain-based, multi-level, location-data-sharing scheme (BMPLS). Location requesters with different access levels can obtain location data with different accuracy, then use Order-Preserving Encryption (OPE) [36] and Merkel tree to verify them. However, BMPLS has the following serious problems: (1) The authenticity of OPE ciphertext, which is used to verify the authenticity of the location data, cannot be verified. (2) Although the OPE scheme [37] used in BMPLS is one-way security [38], and the probability of adversary use of OPE ciphertext to recover the plaintext is negligible, the location requester can use two pairs of border OPE plaintext–ciphertext and the location OPE ciphertext to greatly, illegally, improve the accuracy of received location information. (3) The computation overhead of the location owner in BMPLS is too high.

## 3. Preliminaries

### 3.1. Smart Contract

In the 1990s, Nick Szabo [39] first proposed the concept of smart contract, a computer protocol designed to propagate, verify, and execute contracts. In blockchain technology, especially after Ethereum [40] was proposed, smart contract became an executable program stored in the blockchain. Ethereum provides a platform, a virtual computing environment called Ethereum Virtual Machine (EVM), which allows users to use a complete program-

ming language to write smart contracts. Once certain conditions are triggered, the smart contract program can run automatically. A smart contract allows verified transactions and protocols to be executed between different anonymous parties without a trusted third party. However, a smart contract cannot work actively; it must be called by the blockchain users or other smart contracts.

However, the blockchain is closed, and all external data can only enter the blockchain through transactions. Oracle [41] is the gateway to the outside world for the blockchain, but it is not a specific program, but "a concept". Anything providing external data to the blockchain can be classified as an oracle. When a smart contract concerning extrinsic data is executed, the code then calls for the data from a trusted oracle. In Ethereum, the use of smart contracts uses transactions as the interface. Users deploy or call smart contracts by creating signed transactions, in which the address of the smart contract, the function to be called, and the parameters are written. The transactions are broadcast in the blockchain network, while it is verified. The smart contract is executed locally by the *Miner*s, then the execution result is packaged into the block.

*3.2. Merkle Tree*

A Merkle tree is a hash binary tree, which consists of a root node, a set of intermediate nodes and a set of leaf nodes. The leaf node is the hash value of the stored data, the intermediate node is the hash value of its two child nodes, and the root node is also the hash value of its two child nodes. Any changes in the underlying data (leaf nodes) will be passed up to its parent node, and eventually to the root node. In Bitcoin [29], most nodes use SPV (Simplified Payment Verification) wallets to access the Bitcoin network. Merkle tree-based SPV can be used to verify transactions without storing complete blockchain data.

Users can use a small number of intermediate nodes to generate root nodes. As shown in Figure 1, if the user tries to verify data B, he only needs to obtain four blue intermediate nodes and generate the root node through four hash calculations, then compare the root node with the one in the blockchain. In this paper, we declare that the four blue intermediate nodes in the Merkle tree are the verification data of data B.

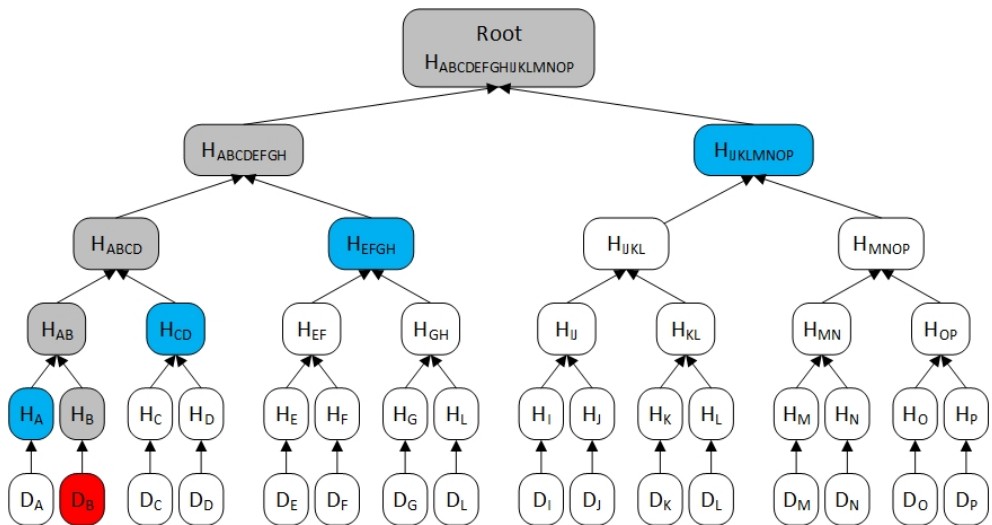

**Figure 1.** Merkle Tree.

## 4. Our Proposed Scheme

*4.1. System Model*

The system model of our proposed scheme is shown in Figure 2.

In addition to the *Miner*s that maintain the blockchain, the proposed blockchain-based, multi-level, location-secure-sharing scheme consists of three entities: *LP*, *LD* and *Fnode*. Any user, if they have sufficient computing and storage capacity, can also act as a *Miner* to maintain the blockchain to gain some benefits.

*LP* is the owner and the provider of location data. *LP* sets the number of access levels $N$ and the accuracy $r = \{r_1, r_2, ..., r_N\}$ according to the demand of *LD*s.

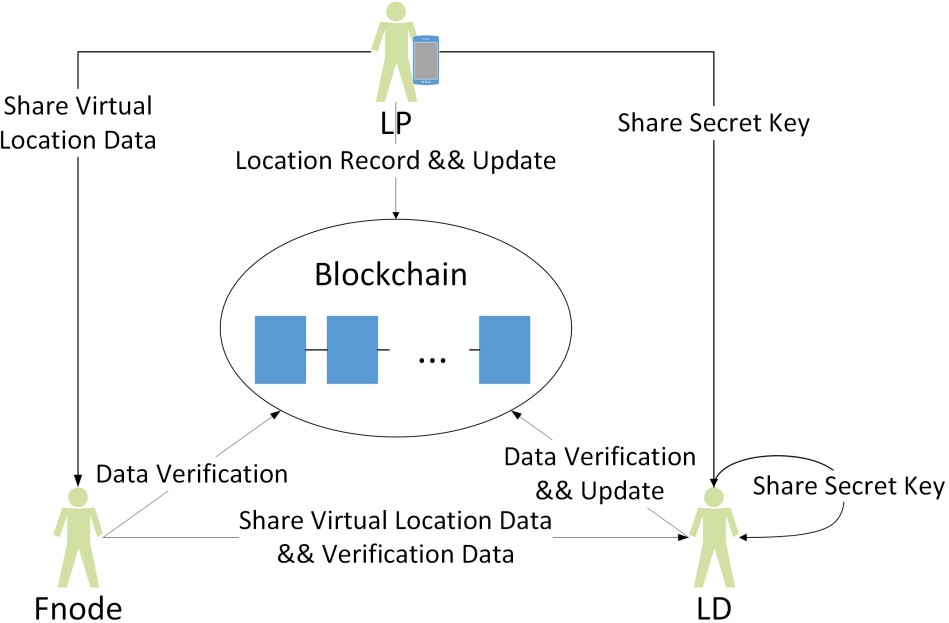

**Figure 2.** Conceptual overview of the proposed scheme.

*LD* is the requester of location data. $LD_i$ denotes the *LD* with level ID $i$, $i \in [0, N]$; the larger the level ID, the lower the access level. $LD_k$ is allowed to determine that the *LP* is located in a circular area with radius $r_k$, $k \in [1, N]$. $LD_0$ is allowed to obtain the exact location of *LP*. Before applying for *LP*'s location data, *LD* calls the smart contract *SC* to record the license for location-sharing into the blockchain. The license contains the addresses (public-key) of *LD* and *LP*, and *LD*'s level ID. *LD* uses the license to obtain the secret key $k_{ud}$ for location updates that *LP* or other *LD*s have received. Therefore, *LD* is not only the data requester, but also the provider. In addition, the license has timeliness. After the block where the license is recorded is mined, the license is only valid within a valid time. With the blockchain, *LD* can verify the received data.

*Fnode* verifies, reserves and provides *LD* the virtual location data from *LP* according to the valid license, which is similar to the queried network node mentioned in Section 8 of [30]. However, in our proposed scheme, only the root node of Merkle tree is recorded in the blockchain, not the data to be verified. *Fnode* can be a network node that intends to gain benefits and be honest but curious.

In our proposed scheme, the smart contract *SC* is designed to do two works: (1) Implement attribute-based access control. *LD* calls *SC* to record a license for location-sharing into the blockchain. (2) Establish an incentive and punishment mechanism. Ensure that nodes can work honestly in the network. We will specifically explain the work of (1) and (2) in Sections 4.3 and 4.4.

### 4.2. Our Proposed Scheme

Our proposed scheme consists of four phases, namely, initialization, location record, location sharing and location update. The following are detailed descriptions of each phase.

#### 4.2.1. Initialization

We designed a new method to set the user's location information to multi-precision, specifically, stripping the location data into multi-level virtual location data and offset vectors. The purpose of this is to enable *LD*s with different access levels to obtain user location information with different precisions. The location offset vector is recorded into

the blockchain in the location update phase, which is illustrated in Section 4.2.4. In the initialization phase, *LP* generates multi-level virtual location data and the secret key for location updates. The specific steps are as follows.

First, *LP*, who is new to the network, executes Algorithm 1, inputting the precision radius set $\{r_1, r_2, ..., r_N\}$, to generate virtual location data $\{Z_0, Z_1, Z_2, ..., Z_N\}$, as shown in Figure 3. $Z_k$ denotes a circle area with $P_k$ as the center point and $r_k$ as the radius, $k \in [1, N]$. The virtual location data meet the following three conditions: (1) The radius length of the circular area $Z_k$ is $r_k$, $k \in [1, N]$. (2) $Z_0$ is located in area $Z_1$. (3) For $a, b \in [1, N]$, if $a > b$, area $Z_a$ completely covers area $Z_b$.

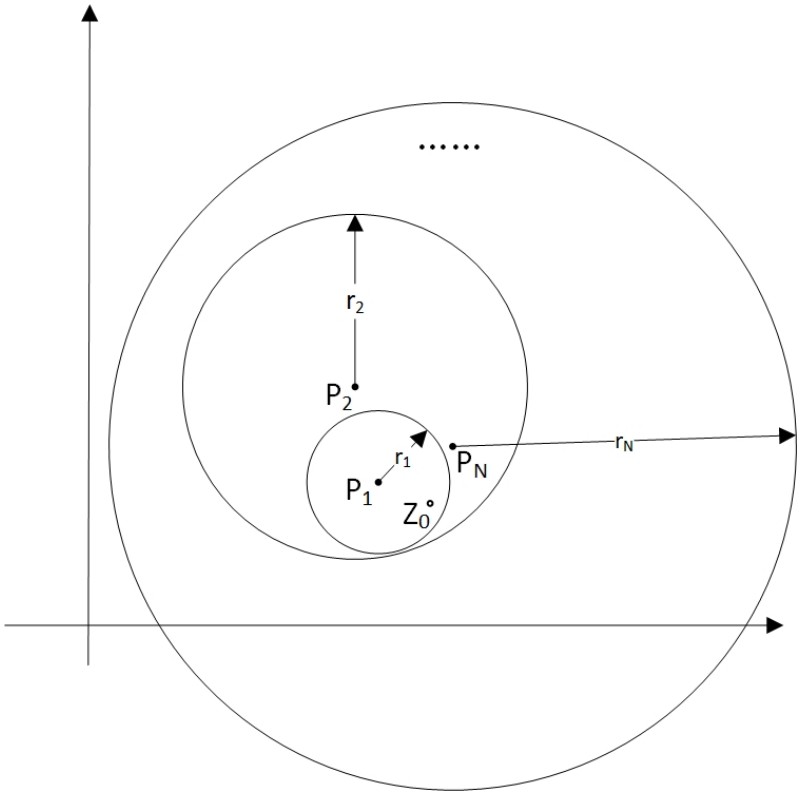

**Figure 3.** Virtual location data.

---

**Algorithm 1** Initialization: pseudocode for randomly generating the virtual location data

---

**Input:** Precision radius set: $\{r_1, r_2...r_N\}$;

1: In the Cartesian coordinates, randomly select a point $Z_0$.;
2: In the circle with $Z_0$ as the center point and $r_1$ as the radius, randomly select a point $P_1$;
3: **for** $i = 1; i \leq N - 1; i + +$ **do**
4:     In the circle with $P_i$ as the center point and $r_{(i+1)} - r_i$ as the radius, randomly select a point $P_{(i+1)}$;
5: **end for**
6: **for** $i = 1; i \leq N; i + +$ **do**
7:     $Z_i \leftarrow P_i || r_i$
8: **end for**
9: **return** $Z_0, Z_1, Z_2, ..., Z_N$;

---

Second, *LP* sends the encrypted virtual location data to *Fnode*s and the encrypted symmetric secret key $k_{ud}^1$ for the 1st location update to *LD*s, which requests the location according to the valid licenses introduced in Section 4.1, the details are shown in Algorithm 2. $Enc()$, $PK_{Fnode}$ and $PK_{LD}$ denote asymmetric encryption function, the public-key of *Fnode*

and the public-key of $LD$. After asymmetric encryption, the data are confidential, even if it is transmitted through an insecure channel.

---

**Algorithm 2** Initialization: pseudocode for sharing virtual location data and key $k_{ud}^1$

---

**Input:** Virtual location data: $Z = \{Z_0, Z_1, ..., Z_N\}$; The secret key for 1-th location update: $k_{ud}^1$;

  1:   $Info_F \leftarrow Enc(PK_{Fnode}, Z)$;

  2:   $Info_D \leftarrow Enc(PK_{LD}, k_{ud}^1)$;

  3:   $LP$ sends $Info_F$ to $Fnode$;

  4:   $LP$ sends $Info_D$ to $LD$;

---

### 4.2.2. Location Record

In the location record phase, $LP$ generates the location record $Record_{LP}$ and records it into the blockchain. The data recorded in the blockchain have immutability. $Fnode$s and $LD$s can verify the data received by the record $Record_{LP}$; the details are shown in Algorithm 2.

In lines 2–3 of Algorithm 3, $LP$ generates the leaf nodes of the Merkle Tree, $Hash()$ denotes the hash function. In line 4, $LP$ builds the Merkle tree $VerMer$ using the leaf notes, $Mer()$ denotes the function to generate a Merkle tree using leaf nodes. In line 8, $LP$ broadcasts $Record_{LP}$ containing the root node $VerMer_{root}$ of $VerMer$ and the hash value of the secret key $k_{ud}^1$ to $Miners$. $Miners$ verify $Record_{LP}$ with $LP$'s digital signature $Sig_{LP}$, then record it into the blockchain through a consensus process, such as proof of work (POW). The consensus mechanism is not a focus of this paper; readers can refer to [15–17].

In lines 11–16 of Algorithm 3, $Fnode$ decrypts the $Info_F$ from $LP$ and generates the Merkle tree $VerMer'$ in the same way as $LP$. $Fnode$ verifies the virtual location data $Z$ by checking whether the root node $VerMer'_{root}$ of $VerMer'$ is equal to the $VerMer_{root}$ in $Record_{LP}$. Further, $Fnode$ verifies whether meets the conditions (2) and (3) mentioned in Section 4.2.1. The incentive and punishment mechanism illustrated in Section 4.4 can prevent $LP$ from cheating $LD$, colluding with less than 50% of $Fnode$s. In lines 19–21, $LD$ decrypts the $Info_D$ from $LP$ and verifies the secret key $k_{ud}^{1'}$ by checking whether the hash value of $k_{ud}^{1'}$ is equal to the $KeyHash$ in $Record_{LP}$.

---

**Algorithm 3** Location record: pseudocode for computing the location record

---

**Input:** Virtual location data: $Z$; The secret key: $k_{ud}^1$;

**Output:** Location record: $Record_{LP}$; Bool varaiable: $Accept_{Fnode}$, $Accept_{LD}$;

  1:   **$LP$ execute:**

  2:   **for** $i = 0; i \leq N; i++$ **do**

  3:      $node_i \leftarrow Hash(i||Z_i)$;

  4:   **end for**

  5:   $VerMer \leftarrow MER(node_0, node_1, ..., node_N)$;

  6:   $KeyHash \leftarrow Hash(k_{ud}^1)$;

  7:   $Info_{Re} \leftarrow VerMer_{root}||KeyHash||Timestamp$;

  8:   $Record_{LP} \leftarrow PK_{LP}||Info_{Re}||Sig_{LP}$;

  9:   $LP$ broadcast $Record_{LP}$ to $Miners$;

10:   **$Fnode$ execute:**

11:   **initialize** $Accept_{Fnode} \leftarrow False$;

12:   $Z' \leftarrow Dec(SK_{Fnode}, Info_F)$;

---

---

**Algorithm 3** *Cont.*

---

13: **for** $i = 0; i \le N; i + +$ **do**
14:     $node_i \leftarrow Hash(i||Z_i')$;
15: **end for**
16: $VerMer' \leftarrow MER(node_0', node_1', ..., node_N')$;
17: **if** $VerMer_{root}' = Record_{LP}.Info_{Re}.VerMer_{root}$ **and** $Z'$ meets the condition (1) and (2) **then**
18:     $Accept_{Fnode} \leftarrow True$;
19: **end if**
20: **return** $Accept_{Fnode}$;
21: *LD* execute:
22: **initialize** $Accept_{LD} \leftarrow False$;
23: $k_{ud}^{1'} \leftarrow Dec(SK_{LD}, Info_{LD})$
24: **if** $Hash(k_{ud}^{1'}) = Record_{LP}.Info_{Re}.KeyHash$ **then**
25:     $Accept_{LD} \leftarrow Ture$;
26: **end if**
27: **return** $Accept_{LD}$;

---

### 4.2.3. Location Sharing

In the location-sharing phase, $LD_n$ uses a valid license to apply for the virtual location $Z_n$ to a *Fnode*. If $LD_n$ has not obtained the secret key $k_{ud}^p$ for location updates, $LD_n$ can use the valid license to apply it to $LD_m$, who has already obtained it. Then, $LD_n$ uses the blockchain to verify the data received. The details are shown in Algorithm 3. In this phase, the data can be securely shared between *Fnode*s and *LD*s without the participation of *LP*.

In lines 2–3 of Algorithm 4, $LD_n$ sends the $Request_n$ containing $Id_{License}$ to a *Fnode*. $Id_{License}$ is the block number of the block where the license is recorded. In lines 5–7, *Fnode* sends the $Info_n^F$ containing the ciphertext of the virtual location data $Z_n$ and the verification data $NODE_n$ to $LD_n$. $NODE_n$ is a set of the intermediate nodes in the Merkle tree, which is the verification data of data $(n||Z_n')$, as introduced in Section 3.2. In lines 9–10, after checking the license, $LD_m$ sends the ciphertext of the secret key $k_{ud}^p$ to $LD_n$, $p$ indicates that *LP* has updated its location for $p - 1$ times at this time. We describe how to update the secret key in Section 4.2.4. In lines 13–14, $LD_n$ decrypts $Loc_n$ of $Info_n^F$ from *Fnode* and $Info_n^m$ from $LD_m$, obtains the virtual location area $Z_n'$ and the secret key $k_{ud}^{p'}$. In lines 15–16, $LD_n$ generates the leaf node $node_n'$, then uses $node_n'$ and $NODE_n$ of $Info_n^F$ to generate the root node $root'$ of *VerMer*, $MerRoot()$ denotes the function to generate the root node of a Merkle tree using two leaf nodes and a set of necessary intermediate nodes. In lines 17, $LD_n$ verifies $Z_n$ by checking whether $root'$ is equal to the $VerMer_{root}$ in $Record_{LP}$. In lines 18–23, $LD_n$ verifies $k_{ud}^{p'}$ by checking whether its hash value is equal to the $KeyHash$ in $Record_{LP}$ or in $Udrec_{LP}^p$. $Udrec_{LP}^p$ is generated and recorded into the blockchain in location update phase, which is illustrated in Section 4.2.4.

### 4.2.4. Location Update

In the location update phase, *LP* generates the location update record $Udrec_{LP}^p$ and records it into the blockchain. *LD* that has obtained the virtual location data and the secret key $k_{ud}^p$ can use $Udrec_{LP}^p$ to update *LP*'s location; the details are shown in Algorithm 5.

In line 2 of Algorithm 5, *LP* translates the geographic coordinate $G^p$, usually expressed in terms of latitude and longitude, into the Cartesian coordinate, where $p$ indicates that *LP* has updated its location for $p - 1$ times at this time. In lines 3–5, *LP* generates and encrypts the offset values $\Delta x^k$ and $\Delta y^k$, $E()$ denotes the symmetric encryption function. In line 6, *LP* generates the hash value of $(k_{ud}^p||Timestamp)$ as the secret key $k_{ud}^{p+1}$ for the next location update. *LP* broadcasts the location update record $Udrec_{LP}^p$ containing the ciphertext of the offset values and the hash value of $k_{ud}^{p+1}$ to the *Miner*s. *Miner*s verify $Record_{LP}$ by *LP*'s digital signature $Sig_{LP}$, then record it into the blockchain.

---

**Algorithm 4** Location sharing: pseudocode for data-sharing and verification

---

**Input:** The block number: $Id_{License}$;

**Output:** Bool varaiable: $Accept_{LD_n}$;

---

1:   $LD_n$ **execute:**

2:   $Request_n \leftarrow PK_{LP}||n||Id_{License}||Sig_{LD_n}$;

3:   $LD_n$ sends $Request_n$ to a $Fnode$ and $LD_m$;

4:   $Fnode$ **execute:**

5:   $Loc_n \leftarrow Enc(PK_{LD_n}, Z_n)$;

6:   $Info_n^F \leftarrow Loc_n||NODE_n$;

7:   $Fnode$ sends $Info_n^F$ to $LD_n$;

8:   $LD_m$ **execute:**

9:   $Info_n^m \leftarrow Enc(PK_{LD_n}, k_{ud}^p)$;

10:   $LD_m$ sends $Info_n^m$ to $LD_n$;

11:   $LD_n$ **execute:**

12:   **initialize** $Accept_{LD_n} \leftarrow False$;

13:   $Z_n' \leftarrow Dec(SK_{LD_n}, Info_n^F.Loc_n)$;

14:   $k_{ud}^{p'} \leftarrow Dec(SK_{LD_n}, Info_n^m)$;

15:   $node_n' \leftarrow Hash(n||Z_n')$;

16:   $root' \leftarrow MerRoot(node_n', Info_n^F.NODE_n)$;

17:   **if** $root' = Record_{LP}.Info_{Re}.VerMer_{root}$ **then**

18:      **if** $p = 1$ **then**

19:         **if** $Hash(k_{ud}^{p'}) = Record_{LP}.Info_{Re}.HashKey$ **then**

20:            $Accept_{LD_n} \leftarrow Ture$;

21:         **end if**

22:      **else**

23:         **if** $Hash(k_{ud}^{p'}) = Udrec^p.Info_{ud}.HashKey$ **then**

24:            $Accept_{LD_n} \leftarrow Ture$;

25:         **end if**

26:      **end if**

27:   **end if**

28:   **return** $Accept_{LD_n}$;

---

In lines 13–14, $LD_n$ decrypts the ciphertext $VE^k$ in the record $Udrec_{LD}^p$, and updates $Z_n$ with the vector $(\Delta x^k, \Delta y^k)$, $D()$ and $Udarea()$ denote the symmetric decryption function and the function to translate location data $Z_n$ using a vector, respectively. In lines 15–17, $LD$ generates $k_{ud}^{p+1}$ and verifies it by checking whether the hash value of it is equal to $KeyHash$ in $Udrec_{LD}^p$. Finally, $LD_n$ can confirm that $LP$ is located in $Z_n^p$. $LD_n$ can translate $Z_n^p$ into the geographic coordinate by $Translate^{-1}()$, which is the inverse function of $Translate()$ in line 2.

In order to prevent users from cheating by sending fake locations, many location-proof schemes [42–46] have been proposed. They focus on using BlueTooth or Wi-Fi technology to verify the authenticity of the user's location point. Here, we can use the location-proof schemes to ensure the authenticity of $VE$ in $Udrec_{LD}^p$.

---

**Algorithm 5** Location update: pseudocode for computing location update record and updating location

---

**Input:** The geographic coordinate of *LP*: $G^p$; The virtual location point of *LP*: $Z_0$; The secret key for p-th location update: $k_{ud}^p$;

**Output:** Bool varaiable: $Accept_{LP_n}$;

1: ***LP* execute:**
2: $P^p = (x^p, y^p) \leftarrow Translate(G^p)$;
3: $\Delta x^p \leftarrow P^p.x^p - Z_0.x_0$;
4: $\Delta y^p \leftarrow P^p.y^p - Z_0.y_0$;
5: $VE \leftarrow E(k_{ud}^p, \Delta x^p || \Delta y^p)$;
6: $k_{ud}^{p+1} \leftarrow Hash(k_{ud}^p || Timestamp)$;
7: $KeyHash \leftarrow Hash(k_{ud}^{p+1})$;
8: $Info_{ud} \leftarrow p || VE || KeyHash || Timestamp$;
9: $Udrec_{LP}^p \leftarrow PK_{LP} || Info_{ud} || Sig_{LP}$;
10: *LP* broadcasts $Udrec_{LP}^p$ to all *Miner*s;
11: ***LP_n* execute:**
12: **initialize** $Accept_{LP_n} \leftarrow False$;
13: $\Delta x^p || \Delta y^p \leftarrow D(k_{ud}^p, Udrec_{LP}^p.VE^p)$;
14: $Z_n^p \leftarrow Udarea(Z_n, (\Delta x^p, \Delta y^p))$;
15: $k_{ud}^{p+1} = Hash(k_{ud}^p || Udrec_{LP}^p.Timestamp)$;
16: **if** $Hash(k_{ud}^{p+1}) = Udrec_{LP}^p.KeyHash$ **then**
17:      $Accept_{LP_n} \leftarrow Ture$
18: **end if**
19: **return** $Accept_{LP_n}$;

---

### 4.3. License Record

As mentioned above, to enable *LD*s with different access levels to obtain user location information with different precisions, we designed a smart contract to implement attribute-based access control. In this section, we illustrate how the smart contract *SC* implements attribute-based access control and records the license. We test *SC* on Remix Solidity IDE, a browser-based IDE that is frequently used to develop Ethereum smart contracts.

As shown in Figure 4, the work of *SC* here is mainly composed of two parts. (1) *LP* and *LD*, respectively, call *SC* to record *LP*'s policy and *LD*'s public attribute. If *LP* has a high-security requirement, *LP* can also record *LD*'s public key as the public attribute and corresponding access level in the policy. However, this will reduce the efficiency of access control and increase the gas consumption of the smart contract. (2) *LD* calls *SC* for *LP*'s location, where the input is the address (public-key) of *LP*. *SC* judges *LD*'s level ID according to *LP*'s policy and *LD*'s public attribute, then records the license into the blockchain. In addition, *LP*s are allowed to call *SC* to modify the policy to change the access level of *LD*s. Note that if the policy is modified, *LP* must re-execute the initialization phase and location record phase for security before updating location, and *LD* also needs to obtain and verify the new virtual location data and the new secret key.

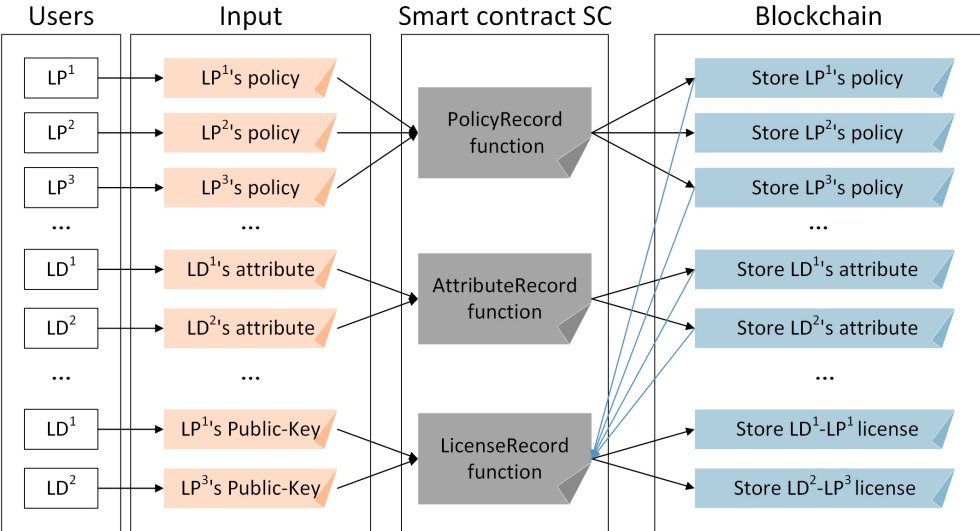

**Figure 4.** Schematic diagram of *SC* recording license.

The smart contract is publicly deployed in the blockchain by the initiator of the system. One can verify the authenticity of the license by checking which smart contract records it.

We simulate the process that *LP* and *LD*, respectively, call *SC* to record the policy and the public attributes, as shown in Figure 5. The policy indicates the access level of a certain type or specific *LD* to *LP*'s location. We also simulate the process by which the *LD*s call *SC* for *LP*'s location; the recorded license is shown in Figure 6.

There is still an unsolved problem of how to ensure the authenticity of *LD*'s recorded public attribute. Our approach is that *LD*s are required to provide relevant proof to an authority who is authorized by the system. After authenticating the identity of the *LD*, the authority confirms the public attribute recorded by *LD* in the smart contract, then the attribute can be recognized. We can specify in the smart contract that some operations can only be executed by the authority, to easily implement the above method.

```
{
"id": "0xEf9590A05A9bA8c87d0eAA57EcFdcED8C14234c4",
"Pub_Atr": "TypeA"
}
...
{
"id": "0xC8864275937d4F949cFb588E78B4Eef058DE95E2",
"Pub_Atr": "TypeZ"
}
...
{
"id": "0x6718765F322ec8160A03EDF4F456fc8b80A0e221",
"Policy":
    [["TypeA","TypeB"],
    ["TypeC"],
    ["0xC8864275937d4F949cFb588E78B4Eef058DE95E2","D"],
    ["TypeE","TypeF"],
    ...
    ["TypeZ"]]
}
```

**Figure 5.** The data formart of *LD*'s public attribute and *LP*'s policy recorded in the blockchain.

```
{
    "from": "0x8c07a2ca270c4a825324f15959d59b0f85de976f",
    "topic": "0x417ee81b26be437c5e304873b5941716e777467ef3266a4c0ccf0c43029272e5",
    "event": "License",
    "args": {
        "LD_Address": "0xEf9590A05A9bA8c87d0eAA57EcFdcED8C14234c4",
        "LP_Address": "0x6718765F322ec8160A03EDF4F456fc8b80A0e221",
        "Level_ID": "0"
    }
}
...
{
    "from": "0x8c07a2ca270c4a825324f15959d59b0f85de976f",
    "topic": "0x417ee81b26be437c5e304873b5941716e777467ef3266a4c0ccf0c43029272e5",
    "event": "License",
    "args": {
        "LD_Address": "0xC8864275937d4F949cFb588E78B4Eef058DE95E2",
        "LP_Address": "0x6718765F322ec8160A03EDF4F456fc8b80A0e221",
        "Level_ID": "2"
    }
}
}
```

**Figure 6.** The data formart of the license recorded in the blockchain.

*4.4. Incentive and Punishment*

In addition to implementing attribute-based access control, in order to drive system users to work honestly, smart-contract SC is also designed to implement an incentive and punishment mechanism, which can be summarized into two parts. First, *Fnode*s can gain benefits from reserving and providing *LP*s' virtual location data for *LD*. Second, *Fnode*s verify *LP*s' virtual location data and report to the network, so as to ensure the authenticity of the location data. Honest *Fnode* obtains benefits; dishonest *LP* and *Fnode* should be punished. The specific method is as follows.

The schematic diagram of the mechanism is shown in Figure 7. When *LP* or *LD*, respectively, calls on *SC* to record policy or license into the blockchain, they must transfer a small fee $E_1$ into *SC*. $E_1$s are distributed to *Fnode*s by *SC* to motivate them to complete their work. In addition to $E_1$, *LP* must transfer a larger amount of the margin into *SC*. If *Fnode* verifies that *LP*'s virtual location data are false, using the method mentioned in line 17 of Algorithm 3, *Fnode* calls *SC* to report *LP*. Within time $T$, if more than 50% of *Fnode*s report *LP*, the collective believes that *LP* provides false data, *LP* is dishonest and *Fnode*s reported that *LP* is honest. Additionally, *LP*'s margin and the fine $E_2$ of the *Fnode*s that have not reported *LP*s will be confiscated and distributed to the *Fnode*s participating in the report. Within time $T$, if fewer than 50% of *Fnode*s report *LP*, the collective believes that *LP* is honest, and the *Fnode*s report that *LP* is dishonest. The fine $E_3$ of *Fnode*s participating in the report will be distributed to the *Fnode*s that have not reported *LP*, and the margin will be returned to *LP*. Note that the margin must be at least greater than the maximum cost of reporting *LP*, which will be discussed further in Section 5.5. In theory, the bigger the margin, the better.

Specifically, the smart contract records a service count variable *Src* and honesty reward variable *Hr* for each *Fnode*. Whenever a new policy is recorded by *LP*, *Src* will increase. We clearly illustrate the change in *Fnode*'s *Hr* in reporting *LP* in Algorithm 6. *Fnode* can redeem incentives in the smart contract based on *Src* and *Hr* at any time. In addition, the smart contract records a timestamp *Ti* for *LP* and a variable *Ih* (the initial value is 1) that marks whether *LP* is honest. If, within time $T$, more than half of *Fnode*s report *LP*, *Ih* is 0, which means *LP* is dishonest. After time $T$, if *Ih* is 1, *LP* can withdraw the margin.

---

**Algorithm 6** Pseudocode for the change in *Fnode*'s *Hr* in reporting *LP*.

---

1: *nFnode*s (contains *Fnode*$_1$) report *LP*;

2: *mFnode*s (contains *Fnode*$_2$) do not report *LP*;

3: **if** Within time *T*, more than 50% of *Fnode*s report *LP* **then**

4:      *Hr* of *Fnode*$_2$ decreases by $E_2$;

5:      *Hr* of *Fnode*$_1$ increases by $(E_2 * m + \text{margin}) / n$;

6: **end if**

7: **if** Within time *T*, less than 50% of *Fnode*s report *LP* **then**

8:      *Hr* of *Fnode*$_1$ decreases by $E_3$;

9:      *Hr* of *Fnode*$_2$ increases by $E_3 * n / m$;

10: **end if**

---

In most practical application scenarios, such as epidemic case statistics and mobile phone application services, it is LP's obligation to provide real data, or the basis for an LP to obtain services. *LP* should not gain a benefit from providing real data. For example, an online travel service provider provides local attraction introductions and different coupons to tourists in different regions. In this case, *LP* provides location data to obtain attraction introductions and coupons.

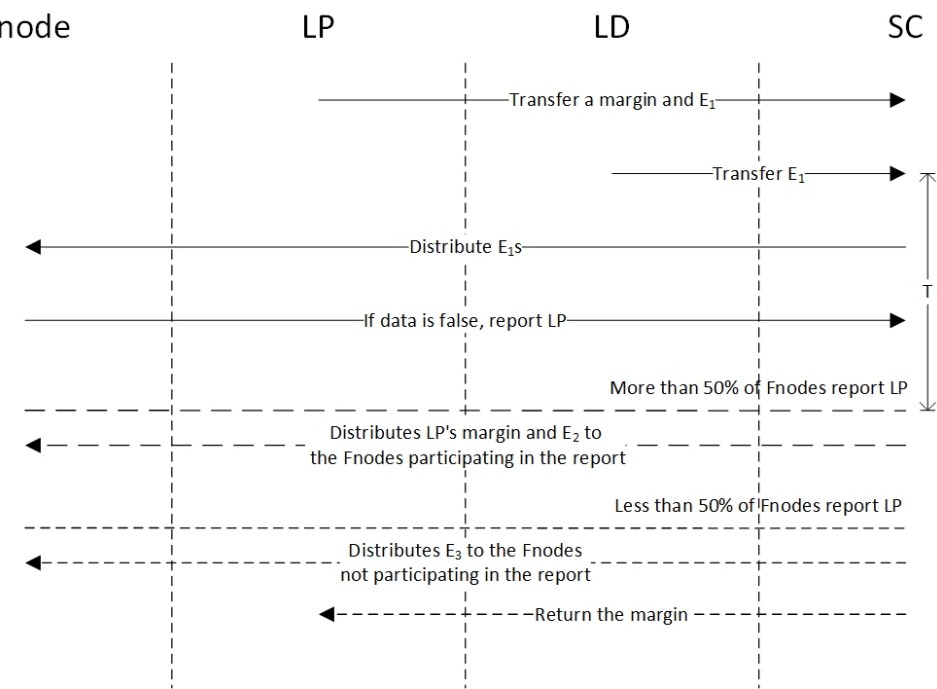

**Figure 7.** Schematic diagram of the incentive and punishment mechanism.

## 5. Experimental Evaluation

In this section, we evaluate the computation overhead of each phase in our proposed scheme and compare it to the existing scheme BMPLS [35]. The experiments are implemented on a PC (CPU: AMD Ryzen 7 4800U CPU @ 1.80 GHz, RAM:16 G, OS: Windows 10) using python-3.7. We used RSA-1024 as the asymmetric encryption algorithm, AES-256 as the symmetric encryption algorithm, SHA-256 as the hash algorithm, and ECDSA-SECP256K1 as the digital signature algorithm in the experiments. The results show that the computation overhead of our proposed scheme is much lower.

In Section 5.5, we deploy our designed *SC* in the Ethereum test network Ganache, whose predecessor was testRPC. Additionally, we evaluate the gas cost of smart contract *SC*.

### 5.1. Computation Overhead of the Initialization Phase

In the initialization phase, *LP* generates and encrypts the virtual location data *Z*, and sends the ciphertext of *Z* to all *Fnode*s. *LP* generates and encrypts the secret key $k_{ud}^1$, and sends the ciphertext of $k_{ud}^1$ to the *LD*s. We evaluate the computation overhead of *LP* in this phase with the different number of *Fnode*s, where we assume that *LP* sends data to 20 *LD*s. Figure 8a shows that the computation overhead of *LP* increases linearly with the number of *Fnode*s. We compare our proposed scheme to BMPLS, as shown in Figure 8b where we use the OPE algorithm proposed in [37]. From Figure 8b we can see that, compared to BMPLS, when *N* is larger than 4, the computation overhead of *LP* in our proposed scheme is much lower; $O(k)$ denotes that the size of OPE plaintext space is $2^k$. The computation overhead in BMPLS increases exponentially with *N*. When *N* is larger, such as $N = 20$, the *LP* in our proposed scheme can still work with a very low computation overhead, while the *LP* in BMPLS will be unable to work because of the extremely high computation burden.

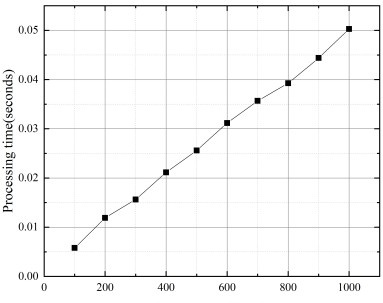

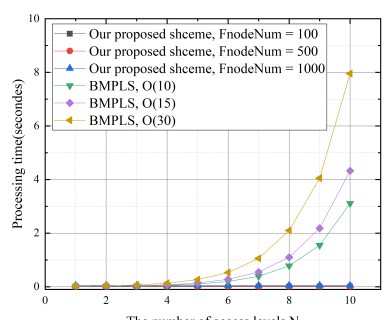

(**a**) The relationship between computation overhead and the number of *Fnode*s.

(**b**) The relationship between computation overhead and the number of access levels *N*.

**Figure 8.** Computation overhead of *LP* in the initialization phase.

### 5.2. Computation Overhead of the Location Record Phase

In the location record phase, *LP* generates the location record $Record_{LP}$. *Fnode*s and the *LP*s verify the data received in the initialization phase with $Record_{LP}$. Figure 9a shows that, compared to BMPLS (here, we use the plaintext-space of OPE $O(15)$ for *LP*), the computation overhead of *LP* in our proposed scheme is much lower, about 13.5% of the computation overhead in BMPLS, where we set $N = 10$. Figure 9b shows that the computation overhead of *Fnode* increases linearly with *N*, and *LD*'s processing time is about 0.004 s.

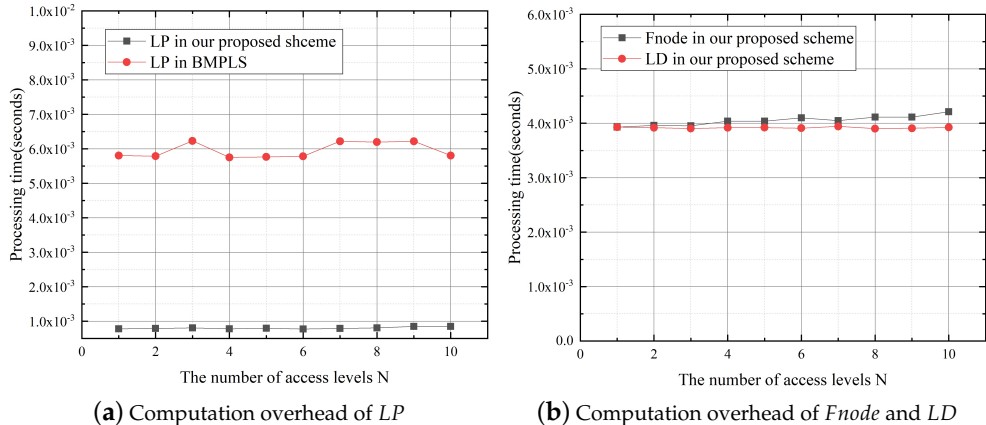

(**a**) Computation overhead of *LP*

(**b**) Computation overhead of *Fnode* and *LD*

**Figure 9.** Computation overhead in the location record phase.

### 5.3. Computation Overhead of the Location Sharing Phase

In the location-sharing phase, *Fnode* and provider-*LD* (the *LD* that has obtained the secret key $k_{ud}$), respectively, provide the virtual location data and $k_{ud}$ to the demander-*LD* (the *LD* that has not obtained the secret key $k_{ud}$). First, we evaluate the computation overhead of data providers including *Fnode* and provider-*LD*. Figure 10a shows that the computation overhead of *Fnode* and provider-*LD* in our proposed scheme is almost equal to that of *LP* in BMPLS. Second, we set $N = 10$ and evaluate the computation overhead of demander-*LD*. Figure 10b shows that the computation overhead of demander-*LD* in our proposed scheme is twice as high as that of *LD* in BMPLS, $LD_0$ and $LD_m$ denote the *LD* that obtains a location point and the *LD* that obtains a location area, respectively. This phase needs to be executed only once during the running of our proposed scheme.

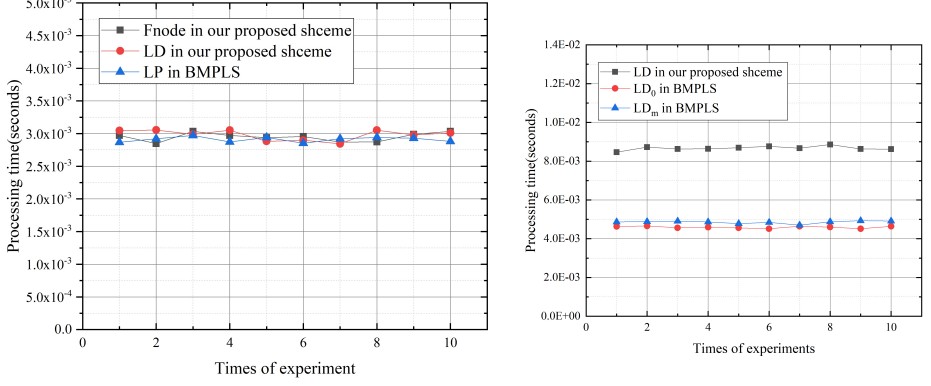

(**a**) Computation overhead of data provider

(**b**) Computation overhead of data demander

**Figure 10.** Computation overhead in the location sharing phase.

### 5.4. Computation Overhead of the Location Update Phase

In the location update phase, a location update record $Udrec_{LP}^{p}$ is generated by *LP* and recorded into the blockchain by *Miner*s to make *LD*s update the *LP*'s location. We set $N = 10$ and evaluate the computation overhead of *LP* and *LD*, respectively. Figure 11 shows that the computation overhead of *LP* and *LD* in our proposed scheme is much lower than that of *LP* and *LD* in BMPLS. The computation overhead of *LP* and *LD* in our proposed scheme is about 9.76% and 0.22% of that in BMPLS, respectively. The reason is that the location-record phase and the location-sharing phase need to be executed again when the *LP* updates location in BMPLS.

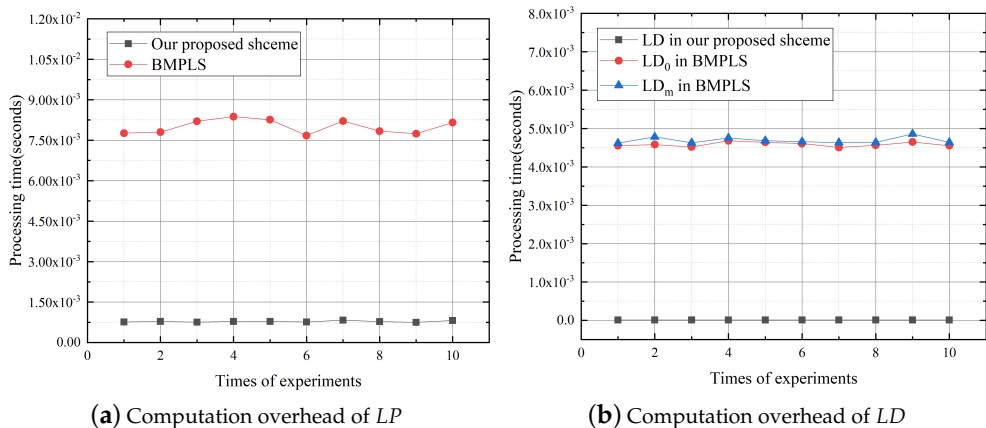

**Figure 11.** Computation overhead in the location update.

We evaluate the computation overhead of *LP* completing all the work with different location update times. Figure 12 shows that the computation overhead in our proposed scheme is significantly lower than that in BMPLS, which is more obvious as the times of location update increase.

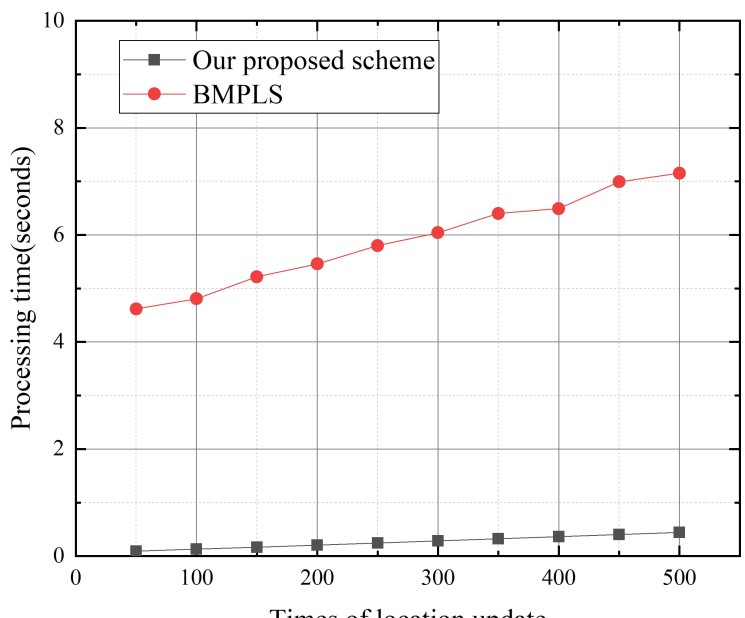

**Figure 12.** Computation overhead of *LP*.

### 5.5. Gas Cost of Smart Contract SC

*SC* is designed to implement attribute-based access control and establish an incentive and punishment mechanism, as introduced in Sections 4.3 and 4.4. First, the gas cost of deploying SC to the blockchain is 1,014,599 gas. The gas cost of *LD* recording the public attribute and the authority confirming it is 44,394 and 42,953 gas, respectively. The gas cost of *LP* recording policy depends on the size of the policy, as shown in Figure 13. This is the gas cost of the first time that *LD* and *LP* record attribute and policy, and the gas cost of modifying is much cheaper. For example, the gas cost of *LP* modifying the attribute to the same one is 19,429 gas. The gas cost of *SC* recording license is 26,382 gas. The gas cost of a *Fnode* reporting *LP* is 49,127 gas.

We noticed that *Fnode* reporting *LP* costs 49,127 gas in Ethereum. This means about $12 at present. This means the margin of *LP* must be greater than the product (maximum gas cost possible) of $12 and the number of *Fnode*s, which would be a large amount. This

may result in our proposed scheme not being put into practice. For this reason, we propose two methods to solve this problem. (1) *Fnode* does not report *LP* in the smart contract, but reports to the authority. The authority records the report result (whether *LP* provides real data) and the dishonest *Fnode*s in the smart contract. In this way, the report in the smart contract is executed only once by the authority. The gas cost of the authority recording increases with the number of dishonest *Fnode*s, as shown in Figure 14. In theory, the margin needs to be greater than the product (maximum gas cost possible) of $1.6 and half of the number of *Fnode*s, which is much lower than before. (2) We used the Ethereum test network discussed above to evaluate the gas cost of the smart contract. Ethereum is one of the hottest blockchain projects, and the cost of the smart contract is high. It is not necessary to use Ethereum in practice. In some cases, system initiators can even create a private chain or alliance chain to implement our proposed scheme, which costs much less. However this is not the focus of this paper.

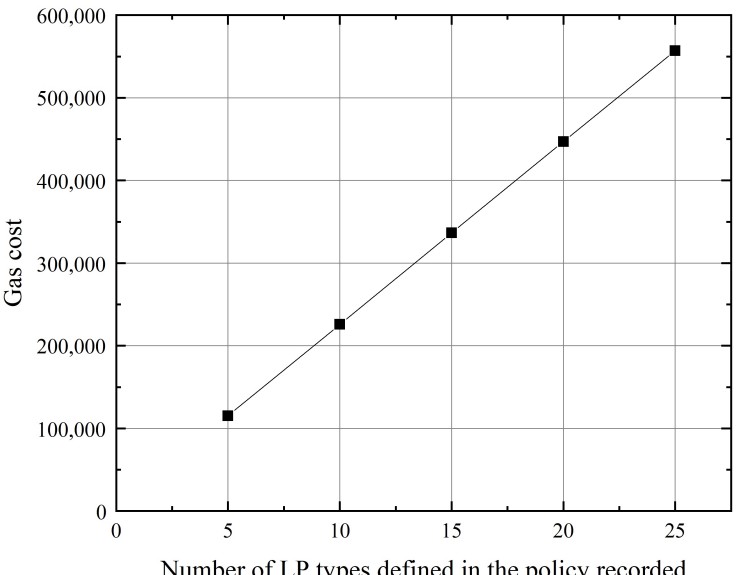

**Figure 13.** Gas cost of *LP* recording policy.

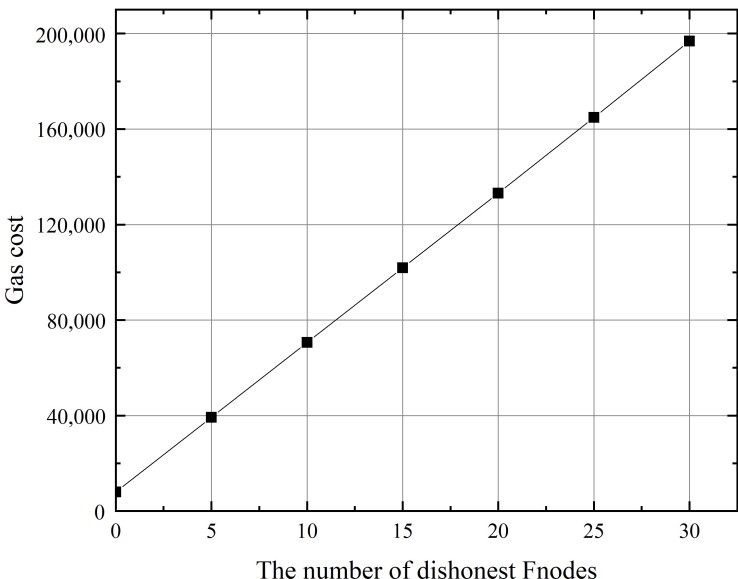

**Figure 14.** Gas cost of the authority recording the report result and the dishonest *Fnode*s.

## 6. Performance of Our Proposed Scheme

In this section, we theoretically analyze the performance of our proposed scheme and prove that, when compared with other related works, our proposed scheme has better performance overall. Specifically, our proposed scheme has the following properties.

### 6.1. Multi-Level Location Privacy Protection and Verifiability

Information security is crucial, but different for the provider and the demander, in an information-sharing process. The provider hopes that data can be shared within the scope of their permission so that privacy can be fully protected, while the requester hopes to verify the authenticity and integrity of the data received.

Multi-level location privacy protection is implemented in our proposed scheme. *Fnode* sends virtual location data with different accuracy to *LD* with different access levels. *LD* cannot illegally obtain location information with higher accuracy. The virtual location data cannot be used to infer the location of *LP*, so that *Fnode* knows nothing about *LP*'s location. Further, because of the key update method, as shown in line 6 of Algorithm 5, our proposed scheme has forward security, and the new *LD* cannot grasp *LP*'s historical location information.

The integrity and authenticity of location data can be verified in our proposed scheme. Because of hash function's collision resistance, *LD* uses hash value and the root node of the Merkle tree to verify the integrity of location data, as shown in lines 15–23 of Algorithm 4. *Fnode* verifies the authenticity of virtual location data for *LD* as shown in lines 12–16 of Algorithm 3. The incentive and punishment mechanism is used to drive users work honestly.

### 6.2. Robust

Only a robust location-sharing scheme can be used in some complex environments, such as medical care, epidemic prevention, IoT device management, etc. We demonstrate that our proposed scheme is robust in two aspects. (1) The robustness of our scheme depends on the robustness of blockchain technology, which is essentially based on the robustness of the consensus mechanism. We choose the common consensus mechanism, POW, which is secure when more than 50% of computation power is held in honest nodes. (2) In our proposed scheme, within the valid time of the license recorded in the blockchain, *LD* can obtain the virtual location data and the secret key for location updates from *Fnode* and the *LD*, respectively, that obtained the key, even if the *LP* is offline. In BMPLS, if *LP* is offline for some reason, no *LD* can obtain *LP*'s location information although its identity is legal, which is unacceptable in some complex environments.

### 6.3. Immutability

Each block in the blockchain, except the genesis block, has a hash pointer that points to the previous block. When there is any change in the parent block, its hash value will change, which forces the "parent block hash value" of the child block to change, resulting in a change in the child block and its hash value. When we use POW as the consensus mechanism, the adversaries need to control at least 51% of the computation power to be able to tamper with the blockchain, the probability of which is negligible. Therefore, the data recorded on the blockchain cannot be tampered with.

### 6.4. Confidentiality

*LP* records $Record_{LP}$ and $Udrec_{LD}^{p}$ into the blockchain. $Record_{LP}$ contains the root node of *VerMer* and the hash value of the secret key $k_{ud}^{1}$. $Udrec_{LD}^{p}$ contains the ciphertext *VE* and the hash value of the secret key $k_{ud}^{p}$. First, due to the hash function's hiding property, the hash values can not reveal any information. Second, as long as the secret key is kept secret, *VE* cannot be decrypted by the adversaries. Therefore, although the blockchain is open to all users in the network, adversaries cannot recover *LP*'s location data from the blockchain.

### 6.5. Decentralization

In the blockchain network, no node is in the central position or has full control and management authority over the blockchain. All nodes store the same blockchain data, which can be freely accessed by users. In addition, we use a smart contract to implement the decentralization of the access control process.

### 6.6. Low Computing Cost

*LP*s that provide location data are often mobile phones or IoT devices that do not have strong computing power. Therefore, the computation overhead in the location sharing scheme must be low. Compared with the related work BMLPS, the computing cost of our proposed scheme is significantly lower, which is proved by the experimental results in Section 5.

### 6.7. Flexibility

Our proposed scheme is flexible, which stems from two aspects: (1) In our proposed scheme, *LP* freely sets the accuracy of the location information obtained by *LD*s with different access levels, which cannot be realized in BMPLS due to using the quad-tree function to partition the region. (2) Access control should be flexible to be adapted to different contexts. In our proposed scheme, *LP* can call the smart contract to record policy to flexibly set the access levels of *LD*s.

### 6.8. Acceptable Gas Cost

In our proposed scheme, the gas costs of the smart contract are acceptable, as shown in Section 5.5. However, frequent modifications to the policy of *LP* and the public attribute of *LD*, and *Fnode* frequently verifying data and reporting *LP*, will bring extra-high gas costs. This situation needs to be avoided. In essence, the more stable the public attribute of *LD*, the better. The addition of new *LD*s should also be controlled.

### 6.9. Performance Comparison

In this section, we compare our proposed scheme with related schemes, including K-anonymity, UB-PPD [32], Medblock [34], and BMPLS [35]. Table 1 shows that our proposed scheme is overall better, where "√", "×" and "-" denote satisfied, dissatisfied and uninvolved, respectively.

**Table 1.** Performance comparison with related works.

| Scheme | K-Anonymity | UB-PPD [32] | MedBlock [34] | BMPLS [35] | Our Proposed Scheme |
|---|---|---|---|---|---|
| Multi-level privacy protection | √ | × | × | × | √ |
| Verifiability | - | × | √ | √ | √ |
| Robust | - | √ | √ | × | √ |
| Immutability | × | √ | √ | √ | √ |
| Confidentiality | √ | × | √ | √ | √ |
| Decentralization | - | √ | √ | √ | √ |
| Low computing cost | √ | √ | √ | × | √ |
| Flexibility | - | × | × | × | √ |
| Acceptable gas cost | - | - | - | - | √ |

## 7. Conclusions

In our proposed scheme, we replaced the central database with a blockchain in the location sharing. The decentralization and immutability of the blockchain make our proposed scheme applicable to a system composed of peer nodes without a central server. Protecting user location privacy and verifiable location data is our focus. We use the

multi-level setting of location data and effective access control to realize the former. To this end, we strip the location data into multi-level virtual location data and offset vectors, and design smart contract *SC* to implement attribute-based access control. The incentive and punishment mechanism established by *SC* can drive users in the network to work honestly. Besides these, we use hash algorithms, Merkle tree and the immutability of blockchain to achieve data verifiability. The results of the experimental evaluation and theoretical analysis show that our proposed scheme is more efficient and secure than the existing works. Smart contracts are not used in related schemes [32,34,35]. In our proposed scheme, the smart contract brings extra gas costs. However, we believe that the gas cost is acceptable when the above properties are achieved.

An increasing number of resources scattered in the network are willing to provide data storage services, which can be seen from the development of technologies such as BT, PT and IPFS in recent years. This type of technology can integrate the resources scattered in different places and work like a cloud server. The main limitation of our proposed scheme is that *Fnode* must not collude with *LD*. *Fnode*, in this paper, can be an node from a BT private network or a DHT network like [32], and it can freely choose whether to provide services for benefit. It is "honest but curious", similar to a cloud data service provider, but much cheaper. In other words, it will not betray the data owner and illegally share information with others. However, it should be prevented from prying into the privacy of the data owner. The system should choose as trustworthy a *Fnode*s as possible, such as those whose social credibility is high.

How to improve the query efficiency in our proposed scheme is a problem that we can continue to study in the future. For *LD*s, maintaining the blockchain is not their original purpose. They tend to be light nodes rather than full nodes, that is, they will not reserve complete blockchain data. Therefore, they query the corresponding blockchain data through other full nodes, similar to SPV-based queries in the bitcoin [29]. It is meaningful to improve the efficiency of this indirect query.

**Author Contributions:** Conceptualization, Q.W. and T.X.; methodology, Q.W. and T.X.; validation, T.X., Y.R. and L.Y.; formal analysis, T.X. and G.M.; investigation, T.X. and G.M.; writing—original draft preparation, Q.W. and T.X.; writing—review and editing, Q.W., L.Y. and G.M.; visualization, T.X.; supervision, Y.R.; project administration, Q.W.; funding acquisition, Q.W. All authors have read and agreed to the published version of the manuscript.

**Funding:** This research was supported by Zhejiang Provincial Natural Science Foundation of China under Grant No. LY19F020039.

**Institutional Review Board Statement:** Not applicable.

**Informed Consent Statement:** Not applicable.

**Data Availability Statement:** Data sharing not applicable.

**Conflicts of Interest:** The authors declare no conflict of interest.

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
