# Peer review of "A New Blockchain-Based Multi-Level Location Secure Sharing Scheme"

_applsci, doi:10.3390/app11052260_

Round 1

Reviewer 1 Report

Dear Authors,

The manuscript is excellent, and well-written overall. Although the results presented in the manuscript seem promising and overall approach is contributing significantly in the body of the literature, I encourage the authors to please consider the attached file minor  suggestions to improvise the presented work more. Thanks

Reviewer 2 Report

## Summary

The authors present a concept for sharing location data on different precision levels using a blockchain. Due to the usage of a distributed architecture they claim security and performance advantages compared with centralised solutions. An incentive and punishment mechanism shall encourage participants to provide honest information. Their solution consists of four kinds of parties: Location providers (LP), location demanders (LD), location verifiers (Fnode), and miners. Finally they present an experimental evaluation for computation overhead and a theoretical analysis of the performance. 

## Critique of the research of the paper

# Strength

The paper explains the idea and algorithms in a very detailed form. The idea of using different precision levels for sharing the location is interesting. Also their encoding in a Merkle Tree and the sharing via blockchain. The experimental evaluation shows that the authors placed a lot of emphasis on scientific work.

# Weakness

Due to the fact that the whole process is explained in a very detailed way the overall idea is hard to understand. Also there are some aspects missing which leads to the confusion of the reader.   

In the opinion of the reviewer there are some aspects that are either problematic aspects or not explained:

- The Fnode is a major weak point in the system because it receives the detailed location information. How is this protected? What differences this from a centralised system?

- The license does not need any consent from LP. Furthermore LP does not know which LD can access and use his/her information. This violates GDPR and is one of the major vulnerabilities of the proposal. There are also no statements how this license can be revoked.

- There are no statements about the frequency LD can retrieve location data. If the frequency is very high position concealment does not make any sense since it can be broken by data correlation. Also using additional data like maps can make the random obfuscation useless.

- It is mentioned that an incentive/penalty mechanism is implemented. The reviewer can only see a penalty mechanism for LP. LP has to transfer a deposit but does not gain anything (but his/her deposit if he/she is honest).

- Form a blockchain perspective the evaluation is completely useless since there is wether statement about the blockchain used for the experiments nor any information about performance. Furthermore, no statement about the costs of this concept are mentioned. Since Gas costs on an Ethereum blockchain can not be neglected the comparison in table 1 is tendentious and incomplete, if not incorrect. There a no information about Gas costs.

# Detailed aspects (referenced by lines)

88ff More existing services should be referenced. Like WhatsApp tracking, Apple tracking, …

143 Here the keyword of an Oracle service has to be mentioned and explained. Reference to the Oracle problem

177 The license is introduced but only explained pages afterwards. At least a cross reference should be given.

204 Which random algorithms are used?

224 LP instead of LD

265 True instead of Ture

277 LDn and LDm. Since it is explained that lower numbers mean higher precision, what is the relationship between n and m? How is ensured that LDm does not provide a precision that LDn is not allowed to see?

How does LDm verifies the license of LDn? 

284 What are verification data NODEn and where do they come from?

393ff It is stated that authenticity of LD is not focus of the paper. The reviewer disagrees strongly since identity of the participants is crucial for the meaningfulness of the whole concept.

404 How does Fnodes verify the location data of LP? What if LP would use a faked GPS location? How can this be detected by Fnode?

## Conclusion

The work might be relevant for the scientific community and the idea is very interesting. In order to be published, the reviewer suggests a strong revision of the paper. The above mentioned aspects should be addressed and explained. 

Round 2

Reviewer 1 Report

Dear Authors,

The manuscript is improved, and much clear now. My all comments are addressed carefully, meanwhile, I would suggest sufficient English improvements in most parts before this article’s publication.

For instance,

  • the location data is set hierarchically and shared with the requester according to the user’s policy
  • the location data is stored/managed hierarchically and shared with the requester according to the user’s policy
  • Finally, we analyze the performances of our proposed scheme and demonstrated that our proposed scheme is overall better than existing schemes.
  • Finally, we analyze the performances of our proposed scheme through simulations and results demonstrated that our proposed scheme is overall better than existing schemes.
  • The problem needs to solve: While effectively sharing location information, the user’s location privacy is protected, and the received data can be true and credible.
  • The problems that need to be solved while effectively sharing location information are the user’s location privacy preservation, and the published data correctness and credibility.

There are several such places where English correction are needed for better readership of this paper.

Reviewer 2 Report

Dear Authors,

Thank you for revising your manuscript and your summary to the revisions.

Please find my comments bellow:

- Thank you for your clarification on Fnode.

- Thank you for taking into account that the license has to be revoked.

- My comment regarding the frequency is explained by your clarification on the Fnode.

- Incentive/penalty: Please elaborate more about this topic, since in my opinion this is crucial. How are incentives be paid? How are deposits be done for the case of punishment? You mention above that Fnode can be trusted - in which cases a punishment of Fnode can happen? How big are those deposits? What is the motivation for LP to participate - except that he/she has to?

Please describe a concrete use case. You mention in your comments that LP has to provide the data. This makes the whole case look in a different way. If LP is forced to provide the data the receiving party can be seen as „trusted third party“.

This whole aspect is described very brief.

- Costs: Please add a paragraph regarding comparison of costs to the 6. and 7.

You mention that Fnode reporting LP costs 49127 gas. This means at the moment $12. Is this realistic? How do you intend to deal with this problem? Changes to the license would be $7. Who would be willing to pay this and why?

If this is not in the focus of your research please explicitly mention that this work is only for academic purpose and will not hold in an productive environment.

Kind regards.

Author Response

请参见附件。

Round 3

Reviewer 2 Report

Thank you for your refinement.